# A Single Tertiary-Care Center Case Series Using Vertical Rectus Abdominis Myocutaneous Flap in the Management of Complex Periprosthetic Joint Infection of the Hip

**DOI:** 10.3390/microorganisms13081962

**Published:** 2025-08-21

**Authors:** Omar Salem, Jing Zhang, George Grammatopoulos, Simon Garceau, Hesham Abdelbary

**Affiliations:** 1Department of Orthopaedic Surgery, The Ottawa Hospital, University of Ottawa, Ottawa, ON K1H 8L6, Canada; ggrammatopoulos@toh.ca (G.G.); sigarceau@toh.ca (S.G.); 2Department of Plastics and Reconstructive Surgery, The Ottawa Hospital, University of Ottawa, Ottawa, ON K1H 8L6, Canada; jzhang1@toh.ca

**Keywords:** periprosthetic joint infection, total hip arthroplasty, vertical rectus abdominus flap

## Abstract

Prosthetic joint infections (PJIs) pose significant challenges, often requiring multiple surgeries that lead to soft tissue loss, dead space, and fibrosis. Wound breakdown increases the risk of polymicrobial infection and treatment failure. The vertical rectus abdominis myocutaneous (VRAM) flap is a proven method for complex wound coverage, but its role in managing hip PJI is underexplored. This study evaluates outcomes of VRAM flap reconstruction in polymicrobial hip PJI. We retrospectively reviewed five patients who underwent VRAM flap reconstruction for polymicrobial hip PJI between December 2020 and December 2023. Primary outcomes included flap survival, infection control, and wound healing. Secondary outcomes were implant retention, postoperative complications, and functional status. At a mean follow-up of 28 months, four patients achieved wound healing and remained infection-free, while one had persistent sinus drainage but retained the implant. Flap survival was 100%, with no necrosis or failure. No major complications requiring reoperation occurred. Two patients developed deep collections, managed with ultrasound-guided drainage (Clavien-Dindo IIIa). Minor complications included donor-site dehiscence (three), flap dehiscence (one), edge necrosis (two), and hernias (two), all managed non-surgically (Clavien-Dindo I/II). All patients retained implants and remained ambulatory. VRAM flap reconstruction is a reliable option for managing complex polymicrobial hip PJI. Flap survival was excellent, and most patients achieved infection control. However, persistent infection and the need for suppressive antibiotics highlight the ongoing challenges in these cases.

## 1. Introduction

PJIs remain one of the leading causes of reoperation following THA, with estimated rates ranging from 1% to 2% after primary arthroplasty, and even higher following revision procedures [1]. PJIs account for approximately 28% of THA revisions, representing significant complications that lead to long-term morbidity and decreased quality of life for affected patients [2].

Surgical management of hip PJI includes debridement, antibiotics, and implant retention, one-stage exchange arthroplasty, and two-stage exchange arthroplasty. While effective, approximately 10% of patients experience persistent infection, with declining success rates after each intervention [3]. Extensive debridement and repeated surgical procedures can lead to large unhealed dead spaces and soft tissue defects, leading to wound dehiscence, chronic drainage, fibrosis, and accumulate blood and tissue fluid, fostering bacterial proliferation. Furthermore, poor vascularity in these areas impairs antibiotic penetration, allowing infections to persist and increasing the risk of further microbial contamination [4,5,6,7,8,9].

Polymicrobial infections constitute a specific subset of PJIs, estimated to account for approximately 4% to 47% of all reported cases [10,11]. These infections pose unique challenges, often associated with lower rates of treatment success, necessitating additional surgical interventions and extended hospital stays [10,11,12].

Muscle flap coverage plays a crucial role in managing persistent PJI with soft tissue defects. Both local and free muscle flaps have been utilized, each with distinct advantages and limitations [9,13]. The VRAM flap is a highly versatile reconstructive option that has been successfully employed to address soft tissue defects in a wide range of anatomical regions [9]. However, its application in managing PJIs of the hip has not been previously described in the literature. This study aims to evaluate the effectiveness of VRAM flaps in managing complex, persistent polymicrobial PJIs of the hip, with a focus on wound healing and infection control following VRAM flap reconstruction. We propose that VRAM flaps offer a superior alternative, combining the coverage benefits of free flaps with the reliability and lower failure rates of local flaps. Their robust vascularity, reduced morbidity, and ability to maintain function make them an effective solution for complex hip PJI cases [14,15,16]. Despite the versatility of VRAM flaps, research investigating their specific safety and outcomes following persistent hip PJIs remains limited.

## 2. Materials and Methods

### 2.1. Study Design and Ethics

This is a retrospective, IRB-approved case series of consecutive patients who underwent VRAM flap reconstruction for hip PJI at our tertiary referral center between December 2020 and December 2023. The primary indication for VRAM flap reconstruction was persistent wound breakdown due to PJI despite prior surgical interventions. Patients who underwent VRAM flap reconstruction for indications other than PJI were excluded. Of the 12 patients initially identified, six were excluded due to tumor-related reconstruction and one due to infection secondary to an intertrochanteric femur fracture with cephalomedullary nail fixation. The final cohort included five patients.

All patients were managed by a multidisciplinary team, including plastic surgeons, orthopedic surgeons, and infectious disease specialists. Data collection included baseline characteristics, comorbidities, number of debridements before and after VRAM flap reconstruction, implant retention at the time of reconstruction and prior flap reconstruction (Table 1 and Table 2).

### 2.2. Patient Classification

Given the complex, multifactorial nature of PJI, patient risk stratification was performed using the McPherson staging system to help contextualize patient risk. This system classifies periprosthetic joint infections based on infection type, systemic host status, and local extremity condition. Infection type is graded as Type I (acute postoperative), Type II (acute hematogenous), or Type III (chronic infection > 4 weeks). Systemic host status is graded as A (uncompromised), B (compromised), or C (significantly compromised) based on comorbidities that impair immune response and healing. Local extremity status is categorized as Grade 1 (uncompromised), Grade 2 (compromised), or Grade 3 (s compromised) based on soft tissue quality, scarring, and vascularity [17] (Table 1).

### 2.3. Outcome

The primary outcome was flap survival and wound healing, defined as the absence of recurrent PJI or wound breakdown at final follow-up. Wound healing was confirmed by complete epithelialization without breakdown, erythema, sinus formation, or fluid collection.

Secondary outcomes included postoperative complications, implant retention, and functional outcome.

Follow-up was defined as the time from VRAM flap reconstruction to the last clinical review. Postoperative complications were categorized using the Clavien-Dindo classification: Grade I (no pharmacologic treatment required), Grade II (pharmacologic treatment required), Grade III (surgical or radiological intervention required, with Grade IIIa not necessitating general anesthesia and Grade IIIb requiring general anesthesia), Grade IV (life-threatening complications), and Grade V (patient death) [18].

### 2.4. Surgical Technique

All procedures were performed in a staged, collaborative fashion between orthopedic and plastic surgery teams under general anesthesia. Patient positioning was dictated by the location of the soft tissue defect and the surgical approach required for debridement and reconstruction. In cases where the posterior approach was utilized, the patient was initially placed in the lateral decubitus position to allow optimal access for extensive debridement and revision hip surgery. Following the orthopedic component of the procedure, these patients were subsequently repositioned into the supine position for flap harvest and inset. Conversely, in patients where the anterior approach was employed, the entire procedure—including both orthopedic and plastic components—was performed with the patient in the supine position, allowing continuous access to both surgical sites without the need for repositioning.

The orthopedic team began with thorough irrigation and debridement, excising all fibrotic scar tissue and obtaining microbiological samples. Hip revision or exchange of modular components was performed when indicated, based on intraoperative findings related to implant stability and infection severity.

Following debridement, the plastic surgeons assessed the hip wound to determine defect size. The patient was then repositioned supine for abdominal exposure. The VRAM flap was designed with a vertical skin paddle, with or without a transverse extension, to ensure adequate coverage and a tension-free inset. A Doppler ultrasound was used to identify major perforators within the abdominal region. A full-thickness skin incision was made, and the deep inferior epigastric artery (DIEA) perforators were carefully isolated, selecting those measuring ≥0.5–1 mm to allow fascia-sparing muscle harvesting. The skin paddle, along with the rectus abdominis muscle, was harvested while preserving most of the rectus sheath to facilitate donor site closure and reduce the risk of abdominal hernia. The muscle’s origin and insertion were released, and the flap was based solely on the DIEA pedicle.

An inguinal incision was made to connect the abdominal and hip wounds, preventing tunneling under the inguinal ligament and minimizing the risk of pedicle compression or twisting. The VRAM flap was rotated 180 degrees into the hip defect, ensuring a tension-free inset and no kinking of the pedicle. Temporary fixation was secured with staples, and perfusion was confirmed using Doppler assessment.

The flap was inset in layers, with Scarpa’s fascia sutured to the thigh fascia using interrupted 0 Maxon sutures, followed by skin closure with 3-0 Monocryl and staples. Two Jackson-Pratt drains were placed beneath the flap. At the donor site, the rectus sheath was repaired with running sutures to prevent abdominal weakness, eliminating the need for mesh reinforcement. Two additional Jackson-Pratt drains were placed at the donor site, and closure was performed in layers with appropriate sutures.

### 2.5. Microbiological Diagnostic Procedures

Intraoperative tissue samples were collected during each debridement surgery, with a minimum of five samples obtained per case, including deep tissue, bone, and prosthetic interfaces. All samples were processed in accordance with institutional microbiology protocols. Culture incubation was carried out for a minimum of 14 days to facilitate the growth of fastidious organisms. Organisms were identified using standard microbiological techniques including Gram staining, biochemical testing, and MALDI-TOF mass spectrometry when necessary.

## 3. Results

### 3.1. Patient Characteristics and Surgical History

Five patients met the inclusion criteria for this study. The cohort included three females and two males, with a mean age of 67.6 years (range: 48–80). All patients had multiple comorbidities, including hypertension, anemia, dyslipidemia, chronic obstructive pulmonary disease (COPD), or prior cerebrovascular accident (CVA). Two patients had a history of smoking or marijuana use. The average body mass index (BMI) was 32.3 kg/m^2^ (range: 24.7–43.4) (Table 1).

According to the McPherson classification, all patients presented with Type III (chronic) infections. Two patients were classified as A2 hosts, reflecting uncompromised systemic health with moderate local compromise, while three were B2 hosts, indicating compromised systemic health and similar local extremity status (Table 1).

The mean number of prior surgical attempts for infection eradication was four (range: 2–8). Four patients did not require further debridement after VRAM flap reconstruction, while one patient underwent five additional debridements and revision surgery due to persistent PJI. Three patients had previously failed wound closure attempts with local flaps—two with gluteus maximus flaps and one with a tensor fascia lata flap. At the time of VRAM flap reconstruction, three patients underwent hip revision or modular component exchange, while the remaining two had only debridement, as their prostheses had already been removed during a prior excision arthroplasty (Table 2).

Wound location and the extent of soft tissue defects varied across the cohort, largely influenced by the site of initial surgical incisions and the extent of subsequent debridements prior to VRAM flap reconstruction. In three patients, the soft tissue loss was predominantly posterior, while in the remaining two, it was anterior (Table 2).

### 3.2. Surgical Outcomes, Complications, and Functional Outcome

The mean follow-up period after VRAM flap reconstruction was 28 months (range: 12–47 months). At final follow-up, flap survival was 100%, with no cases of necrosis or failure. Four patients achieved complete wound healing and remained infection-free, while one patient experienced persistent sinus discharge but retained her implant.

All patients had retained implants at final follow-up, either through initial implant retention or successful reimplantation (Table 3). Two patients had their implants removed prior to VRAM flap reconstruction as part of infection control (Figure 1). Following reconstruction, both achieved successful infection eradication and proceeded to prosthesis reimplantation. The remaining three patients retained their implants at the time of VRAM flap reconstruction and underwent extensive debridement with modular component exchange (Figure 2). Of these, one achieved complete wound healing and infection control. The other two experienced persistent infection—one required flap elevation, five additional debridements, and a two-stage revision. Despite the prolonged course, infection was ultimately eradicated, and reimplantation was performed 2 years and 8 months later, with the flap remaining viable throughout. The second patient continued to have a draining sinus but retained the implant. A summary of procedures and outcomes is shown in Figure 3.

Postoperative complications are summarized using the Clavien-Dindo classification. Patient 1 developed graft edge necrosis, donor site dehiscence, ventral hernia, and seroma (Grade IIIa). Patient 2 developed a ventral hernia (Grade IIIa). Patient 3 experienced graft edge necrosis and donor site dehiscence (Grade IIIa). Patient 4 had donor site dehiscence (Grade I), and Patient 5 experienced flap site dehiscence (Grade I). No life-threatening complications or deaths occurred. These findings underscore the expected morbidity associated with VRAM harvest but also highlight the absence of flap loss, major surgical reinterventions, or life-threatening events (Table 3).

All patients were ambulatory at final follow-up, with four requiring assistive devices. Three of the five patients were placed on long-term suppressive oral antibiotic therapy (Table 3).

### 3.3. Microbiological Findings and Antibiotic Therapy

Infection characteristics varied across the cohort. Three patients initially presented with monomicrobial infections that progressed to polymicrobial profiles, while two patients had polymicrobial PJIs from the outset. A wide array of organisms were identified over the course of multiple surgeries, including Gram-negative bacilli (*Morganella morganii*, *Pseudomonas aeruginosa*, *Klebsiella pneumoniae*, *Citrobacter freundii complex*, *Serratia marcescens*), Gram-positive cocci (*Enterococcus faecalis*, vancomycin-resistant *Enterococcus faecium*, *Staphylococcus epidermidis*, coagulase-negative *Staphylococcus aureus*), and fungi (*Candida albicans*). *Corynebacterium striatum* and *coryneform bacteria* were also identified in several cases.

Detailed resistance profiles were obtained, revealing resistance to common *beta-lactams*, *fluoroquinolones*, *TMP/SMX*, and *vancomycin* in selected organisms. Notably, vancomycin-resistant *enterococci* (VRE) were identified in three patients, necessitating use of agents such as *daptomycin* and *linezolid*. *Ciprofloxacin* resistance was also frequent, particularly among *Corynebacterium striatum* and *Staphylococcus epidermidis* isolates.

Antibiotic regimens were adjusted per culture and sensitivity results. Initial empiric therapies commonly included *cefazolin*, *vancomycin*, and *piperacillin-tazobactam*. Targeted regimens included *meropenem*, *daptomycin*, *ciprofloxacin*, *metronidazole*, and *linezolid*. Suppressive antibiotic therapy was used in three patients, extending for weeks to months depending on organism sensitivity and implant status (Table 4).

## 4. Discussion

Managing persistent polymicrobial PJIs is notoriously challenging, with lower treatment success rates [10,19]. Effective treatment requires a multidisciplinary approach considering patient health, implant stability, pathogen virulence, and the availability of suppressive antibiotics. These cases often necessitate multiple surgical interventions, extended hospital stays, and prolonged antimicrobial therapy, emphasizing the importance of multidisciplinary PJI meetings for optimizing treatment strategies [10,19,20].

Soft tissue defects further complicate PJI management, especially when infections extend deeply, involve exposed bone or prosthetic components, or remain in direct communication with the external environment [21]. The efficacy of open wound management, such as frequent dressing changes or vacuum-assisted negative pressure wound therapy, remains debated, as prolonged open wounds have been associated with conversion from monomicrobial to polymicrobial infections. Fröschen et al. found that 65% of patients with prolonged open wounds experienced severe complications, including Girdlestone procedures, fistula formation, amputation, or death [22]. While, Valenzuela et al. reported that 34% of monomicrobial PJIs progressed to polymicrobial infections when treated with open wound management for over two weeks [11]. These findings reinforce the importance of early soft tissue closure to reduce infection-related morbidity.

Muscle flaps offer a robust solution for PJI-related soft tissue defects by allowing radical debridement, reducing tension, and providing well-vascularized tissue that fills dead space, that promotes neovascularization and antibiotic penetration [14,15,16]. Various local muscle flaps, including the gluteus maximus, tensor fascia lata, vastus lateralis, and rectus femoris, have been used for hip PJI reconstruction, while extensive defects may require free tissue transfers, such as the latissimus dorsi flap. Local flaps are advantageous due to their simpler surgical technique, lower failure rates, and immediate availability, but they offer limited tissue volume and may lead to lower extremity weakness and increased local morbidity. Free muscle flaps such as the latissimus dorsi, provide well-vascularized coverage, but their reliance on microvascular anastomosis increases surgical complexity and failure rates [9,10,11,12,13,14,15,16,17,18,19,20,21,22,23,24].

The rectus abdominis muscle flap was first described for chronic hip infections in 1983 by Irons, who successfully treated three patients with deep cavities, soft tissue defects, and ongoing drainage, including one requiring a Girdlestone procedure [24]. Irons’ technique involved releasing the muscle’s proximal attachment and passing it through a defect in the acetabulum. Subsequent refinements included releasing both the proximal and distal muscle insertions, allowing rotation of the flap around the deep inferior epigastric artery (DIEA) [23,24,25,26]. The Mathes and Nahai type III VRAM flap, with dual perfusion from the deep superior epigastric artery and DIEA, is widely used in reconstructive surgery due to its reliable vascular anatomy, ease of dissection, and minimal donor-site morbidity [14,16]. Studies confirm its safety, even in high-risk patients with vascular disease or prior radiation therapy [14,15,16], and superior outcomes compared to thigh-based flaps in pelvic and perineal reconstruction [27].

VRAM flaps offer the benefits of both local and free muscle flaps, providing ample skin and muscle for tension-free closure while ensuring robust vascularity through the DIEA without microvascular anastomosis. Unlike local flaps, they do not increase local morbidity such as fibrosis, atrophy, and chronic infection, thereby preserving lower extremity strength and minimizing scarring—critical for future implant retention or reimplantation.

In our study, all patients had undergone an average of four prior PJI surgeries and prolonged antibiotic therapy, with infection control achieved only after definitive soft tissue reconstruction using a VRAM flap (Table 2). All flaps remained viable, leading to wound closure with no flap failures or major complications requiring further surgery. Notably, flap viability was preserved even during subsequent debridement and reimplantation. While one patient developed a persistent draining sinus, this may be attributable to implant retention with change in the modular components. The remaining patients underwent either one-stage or two-stage revision, consistent with literature favoring this approach for better infection control [10,11,12].

In our cohort, the majority of patients were Type III infections with compromised systemic (B) and local (Grade 2) status, reflecting the high-risk nature of this population and underscoring the need for durable soft tissue reconstruction. Given the complex, multifactorial nature of PJI treatment failure—including repeated debridement, prolonged antibiotic therapy, and local tissue compromise—this system provides a valuable framework to contextualize patient risk and the challenges encountered prior to VRAM flap reconstruction.

An important technical finding of this study was the flap’s adaptability. Despite varied soft tissue defect locations and differing surgical approaches—posterior in three cases and anterior in two—the VRAM flap consistently achieved tension-free, well-perfused inset. Its long arc of rotation allowed it to reach posterior defects, traditionally considered more difficult to cover and often reserved for free flap reconstruction. This reinforces the VRAM flap’s utility in revision hip arthroplasty, especially when posterior exposure is required.

All cases in this series progressed to polymicrobial infections. While two patients initially had polymicrobial PJIs, additional pathogens were identified during subsequent debridements, likely due to contamination from chronic open wounds. This aligns with prior studies reporting increased pathogen diversity in cases with prolonged wound exposure. Marculescu and Cantey found that PJI patients with soft tissue defects or wound drainage were significantly more likely to develop polymicrobial infections [12]. Other studies on hip wounds treated with muscle flaps similarly reported high polymicrobial infection rates (Table 5).

All patients underwent multiple debridements, with organisms and sensitivities changing over time. This highlights the dynamic nature of PJI microbiology and the importance of repeated sampling and microbiological surveillance throughout treatment. The microbiological profiles underscore the progressive nature of these infections. The detailed microbiological findings reported in this study contribute valuable insights and offer a robust platform for future investigations targeting biofilm-associated resistance, multidrug pathogen emergence, and tailored antimicrobial protocols. While surgical treatment remains the cornerstone of management, integration of microbiological data is critical for treatment success.

Although most studies report favorable healing rates with the use of muscle flaps in hip PJIs, prosthesis salvage rates remain low (Table 5). Furthermore, limited data exist regarding long-term functional outcomes and the morbidity associated with muscle flaps [9]. Given the complexity of these cases, predicting functional deficits post-infection clearance remains difficult. Suda et al. emphasized the need for standardized functional outcome scoring in these patients [28]. Some authors suggest that mobility and recovery are best preserved in patients who retain their implants [5,6,7,8]. In this study, all patients retained their implants and maintained adequate mobility following reimplantation.

Two patients developed ventral hernias, none of which required surgical repair during the follow-up period. Three patients experienced graft edge necrosis, and four had donor-site dehiscence. One patient developed a seroma. All complications were managed non-operatively and did not exceed Clavien-Dindo Grade IIIa. Importantly, these events did not result in flap failure, additional surgery, or long-term loss of mobility. Still, future studies should further investigate long-term donor-site outcomes.

Limitations of this study include its retrospective design, small sample size, and relatively short follow-up period. Further prospective studies with longer follow-up are necessary to evaluate outcomes comprehensively, particularly with regard to functional status, quality of life, and donor-site morbidity.

**Table 5 microorganisms-13-01962-t005:** Studies on muscle flaps for treatment of hip PJI.

Studies	No. of Patients	Flap Type	No. of Polymicrobial	Average Prior PJI Surgeries	Wound Condition	Implant Retention
Arnold, 1983 [21]	7	6 RF, 3 VL	7 (78%)	10.3 (6–25)	Healed	0 (0%)
Jones, 1991 [23]	5	3 RA, 1 LD, 1 VL	NA	(4–32)	Healed	1 (20%)
Meland, 1991 [13]	27	23 RF, 8 VL, 1 TFL	27 (84%)	4.2 (1–21)	Healed	18 (67%)
Lewis, 1994 [29]	1	TFL	NA	4	Healed	0 (0%)
Windle, 1996 [26]	3	RA	NA	Multiple	Healed	3 (100%)
Lee, 1996 [6]	7	VL	2 (29%)	2.9 (2–5)	Healed	0 (0%)
Ross, 1998 [25]	1	RA	NA	4	Healed	0 (0%)
Ikeda, 2001 [30]	1	VL	0	NA	Healed	0 (0%)
Gusenoff, 2002 [20]	4	VL	NA	2.6 (1–5)	Healed	3 (75%)
Huang, 2005 [31]	4	VL	2 (50%)	6 (4–8)	Healed	2 (50%)
Shieh, 2007 [8]	1	VL	0	multiple	Healed	1 (100%)
D’Ettorre, 2010 [7]	2	VL	0	NA	Healed	2 (100%)
Suda, 2010 [28]	119	VL	6 (5%)	4.9 (2–25)	Healed	4 (3%)
Choa, 2011 [5]	24	20 RF, 5 VL	22 (88%)	4 (1–8)	22 healed, 2 sinuses	14 (58%)
Ricciardi, 2017 [4]	4	GM	4 (100%)	4.75 (2–8)	Healed	3 (75%)

Abbreviations: RF: Rectus Femoris, VL: Vastus Lateralis, LD: Latissimus Dorsi, RA: Rectus Abdominus, GM: Gluteus Maximus, NA: not available.

## 5. Conclusions

This study demonstrates the efficacy of the VRAM flap as a reliable and versatile option for managing complex polymicrobial PJIs of the hip. The use of VRAM flaps successfully facilitated infection control, implant viability, and ensured complete wound healing in the majority of patients, with minimal complications. Although challenges such as polymicrobial infections and the need for long-term suppressive antibiotics were encountered, VRAM flap reconstruction provided a robust solution to soft tissue defects, preserving limb function and facilitating reimplantation. While the results are promising, further prospective studies with larger sample sizes and longer follow-up are needed to fully assess the long-term functional outcomes and potential risks associated with VRAM flap use in this context.

## Figures and Tables

**Figure 1 microorganisms-13-01962-f001:**
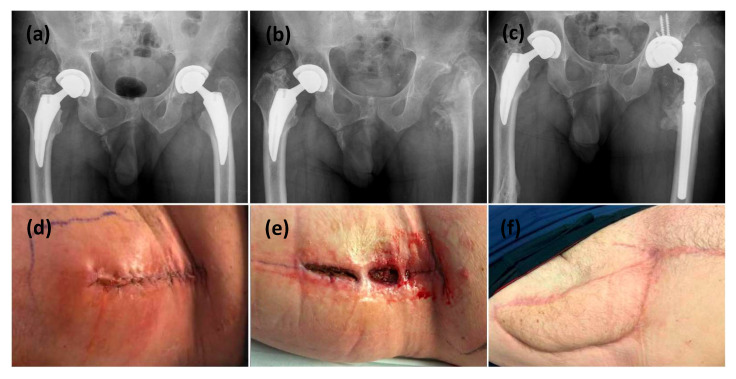
This is a figure of left hip PJI with all revisions performed through a direct anterior surgical approach. (**a**) Initial AP pelvis x-ray showing bilateral osteoarthritis necessitating bilateral uncemented total hip replacements. (**d**) clinical presentation with initial symptoms of PJI at 4 weeks post index surgery. (**b**) After 3 operations to control PJI, the patient ended up with a resection arthroplasty. (**e**) The hip incision failed to heal after 4 weeks. (**f**) a VRAM flap was used to reconstruct the anterior hip wound and a reimplantation for the prosthesis was performed through an anteriorapproach. (**c**) Final postoperative radiograph showing the definitive implant inserted after wound healing and infection controlled.

**Figure 2 microorganisms-13-01962-f002:**
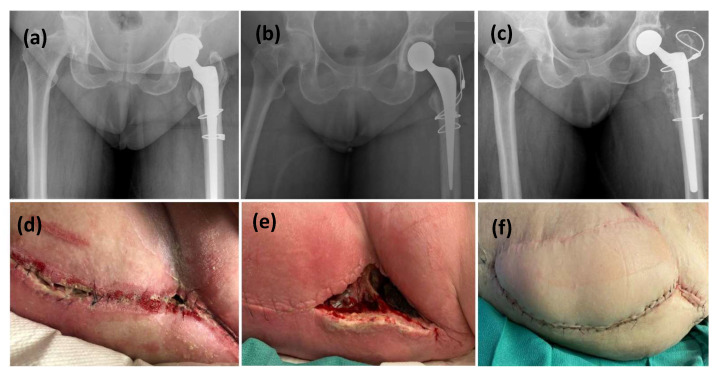
This is a figure of left hip PJI with all revisions performed through a posterior surgical approach. (**a**,**d**) radiographic and clinical presentation with initial symptoms of PJI, (**b**) The patient underwent a stage-1 revision with an antibiotic-cement spacer, (**e**) wound dehiscence at 4 weeks post spacer implantation, (**c**,**f**) The spacer was then revised to Stage-2 definitive implant with a VRAM flap at the same time.

**Figure 3 microorganisms-13-01962-f003:**
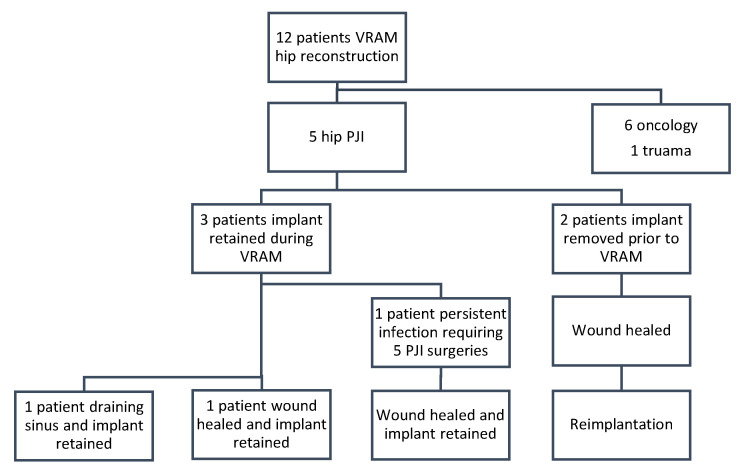
Flow diagram to represent patient procedures and outcomes for infection clearance and implant retention.

**Table 1 microorganisms-13-01962-t001:** Demographic features.

Patient	Age	Sex	Ethnicity	Site	BMI	Comorbidities	Smoking	ASA	McPherson Staging System		
									Infection Type	Systemic Host Grade	Local Extremity Grade
1	80	F	African	R	31.4	HTN, DSLP, CVA, Anemia, GERD	Never	4	III	B	2
2	78	F	White	R	24.7	HTN, DSLP, Anemia, Mitral and tricuspid regurgitation, osteoporosis	Never	3	III	A	2
3	65	F	White	L	43.4	HTN, GERD, HNPCC, COPD, Anemia,	Former	3	III	A	2
4	67	M	White	L	30.43	HTN, CVA, Adrenal gland hypofunction, Hiatal hernia, Asthma, Pseudocholinesterase deficiency.	Marijuana	3	III	B	2
5	48	M	White	L	31.56	HTN, GERD	Marijuana	3	III	B	2

Abbreviations: M: male, F: female, HTN: hypertension, DSLP: dyslipidemia, GERD: gastroesophageal reflux disease, HNPCC: Hereditary nonpolyposis colorectal cancer, COPD: chronic obstructive pulmonary disease, CVA: cerebrovascular accident.

**Table 2 microorganisms-13-01962-t002:** Surgical history.

Patient	Surgical Approach	PJI Surgeries Before VRAM	PJI Surgeries After VRAM	Prior Flap Recon	Implant During VRAM
1	Posterior	2	0	No	Retained
2	Posterior	5	0	GM	Retained
3	Anterior	3	5	GM	Retained
4	Posterior	8	0	TFL	Removed
5	Anterior	3	0	No	Removed

Abbreviations: GM: Gluteus maximus, TFL: Tensor fascia lata.

**Table 3 microorganisms-13-01962-t003:** Complications and Outcome at final follow-up.

Patient	Follow Up Duration	Complication	Clavien-Dindo Classification	Flap Condition	Wound Condition	Chronic Suppression	Implant	Function
1	24 M	Graft edge necrosis, Donor site dehiscence, Ventral hernia, seroma	II, IIIa	Viable	Draining sinus	none	Retained	Ambulating with walker
2	12 M	Ventral hernia	I	Viable	Healed	Abx	Retained	Ambulating with walker
3	47 M	Graft edge necrosis, Donor site dehiscence	II	Viable	Healed	none	Retained	Ambulating with walker
4	40 M	Donor site dehiscence.	II, IIIa	Viable	Healed	Abx	Retained	Ambulating with walker
5	17 M	Graft site dehiscence	II	Viable	Healed	Abx	Retained	Independent ambulation

Abbreviations: M: months, Abx: antibiotics.

**Table 4 microorganisms-13-01962-t004:** Microbiological Profile, Resistance/Susceptibility, and Antibiotic Treatment.

Patient	PJI Surgeries	Organism(s) Identified	Resistance/Susceptibility Profile	Antibiotic Treatment
1	1	*Morganella morganii*	R—*Amoxicillin/Clavulanate*, *Ampicillin*, *Cefazolin*, *Ceftriaxone* S—*Ciprofloxacin*, *Gentamicin*, *Piperacillin/Tazobactam*, *TMP/SMX*, *Tobramycin*	*Piperacillin/Tazobactam*, *Meropenem*
		*Pseudomonas aeruginosa*	S—*Ceftazidime*, *Meropenem*, *Piperacillin/Tazobactam*, *TMP/SMX*, *Tobramycin*	
	2	Vancomycin-resistant *Enterococcus* (VRE)	R—*Ampicillin*, *Tetracycline*, *Vancomycin* S—*Daptomycin*, *Linezolid*	*Daptomycin*, *Meropenem*
	VRAM	*Corynebacterium striatum*	R—*Ciprofloxacin*, *Erythromycin*, *Penicillin*, *Tetracycline* I—*Gentamicin* S—*Linezolid*, *Vancomycin*	*Meropenem*, *Daptomycin*
2	1	No growth	N/A	*Cefazolin*
	2	No growth	N/A	*Cefazolin*, *Vancomycin*
	3	No growth	N/A	*Vancomycin*, *Ceftriaxone*
	4	Coagulase-negative *Staph aureus*	S—*Cefazolin*, *Cloxacillin*, *Rifampin*, *TMP/SMX*	*Cefazolin*
	5	*Staph. epidermidis*	R—*Cefazolin*, *Ciprofloxacin*, *Clindamycin*, *Eloxacillin*, *erythromycin*, *Levofloxacin*, *Tetracycline*, *TMP/SMX* S—*Rifampin*, *Vancomycin*	*Daptomycin*, *Doxycycline*
	VRAM	*Staph. epidermidis*	R—*Cefazolin*, *Cloxacillin*, *TMP/SMX* S—*Rifampin*, *Vancomycin*	*Ciprofloxacin*, *Daptomycin*, *Rifampin*
3	1	*Enterococcus faecalis*	S—*Ampicillin*	*Ampicillin*, *Metronidazole*
		*Prevotella bivia*	S—*Metronidazole*, *Amoxicillin/Clavulanate*, *Piperacillin/Tazobactam*, *Carbapenems*	
	2	*Klebsiella pneumoniae*	R—*Ampicillin* S—*Ciprofloxacin*, *Gentamycin*, *Cefazolin*, *TMP/SMX*, *Tobramycin*	*Ampicillin*, *Meropenem*
		*Pseudomonas aeruginosa*	S—*Ceftazidime*, *Ciprofloxacin*, *Gentamycin*, *Piperacillin/Tazobactam*, *Meropenem*, *Tobramycin*	
		*Citrobacter freundii complex*	R—*Amoxicillin/Clavulanate*, *Ampicillin*, *Cefazolin*, *Ceftriaxone* S—*Ciprofloxacin*, *Gentamycin*, *Piperacillin/Tazobactam*, *TMP/SMX*, *Tobramycin*	
	3	Same as above		
	VRAM	*Coryneform bacteria*	Not specified	*Ampicillin*, *Meropenem*
	1	*Corynebacterium striatum*	R—*Ciprofloxacin*, *Erythromycin*, *Gentamicin*, *Penicillin*, *TMP/SMX* S—*Linezolid*, *Vancomycin*	*Ciprofloxacin*
	2	*Corynebacterium striatum*	Same as above	*Ciprofloxacin*, *Meropenem*
	3	*Candida albicans*	S—*Fluconazole*	*Ciprofloxacin*, *Fluconazole*
	4	*Serratia marcescens*	R—*Amoxicillin/Clavulanate*, *Ampicillin*, *Cefazolin*, *Ceftriaxone*, *Ciprofloxacin.* I—*TMP/SMX* S—*Gentamycin*, *Meropenem*, *Pipercillin/Tazobactam*, *Tobramycin*	*Meropenem*, *Fluconazole*
		*VRE faecium*	S—*Ampicillin*	
		*Candida albicans*		
	5	*VRE faecium*	S—*Ampicillin*	*Meropenem*, *Fluconazole*
4	1	*MRSA*	No sensitivity data (external facility)	*Vancomycin*
	2	*MRSA*, *Enterococcus faecalis*	No sensitivity data (external facility)	*Vancomycin*, *Ertapenem*
	3	*MRSA*, *Enterococcus faecalis*, *Serratia*	No sensitivity data (external facility)	*Vancomycin*, *Ertapenem*, *Daptomycin*
	4	*Candida albicans*	Not specified	*Meropenem*, *Daptomycin*
	5	*Candida albicans*	Not specified	*Meropenem*, *Daptomycin*, *Fluconazole*
	6	No growth	N/A	*Meropenem*, *Daptomycin*, *Fluconazole*
	7	No growth	N/A	*Meropenem*, *Daptomycin*, *Fluconazole*
	8	No growth	N/A	*Meropenem*, *Daptomycin*, *Fluconazole*
	VRAM	No growth	N/A	*Moxifloxacin*, *Daptomycin*
5	1	*Pseudomonas aeruginosa*	S—*Ceftriaxone*, *Ciprofloxacin*, *Gentamicin*, *Meropenem*, *Piperacillin/Tazobactam*, *Tobramycin*	*Ceftazidime*, *Ciprofloxacin*
	2	No growth	N/A	*Ceftazidime*, *Ciprofloxacin*
	3	*VRE faecium*	R—*Vancomycin*, *Ampicillin* S—*Daptomycin*, *Linezolid*	*Daptomycin*, *Ciprofloxacin*
	VRAM	No growth	N/A	*Daptomycin*, *Ciprofloxacin*

Abbreviations: S: susceptible, R: resistant, I: intermediate.

## Data Availability

The data presented in this study are available on request from the corresponding author upon reasonable request. Due to the retrospective nature of the study and the inclusion of patient-specific clinical data, access is restricted to protect patient confidentiality.

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
