# Peer review of "A Single Tertiary-Care Center Case Series Using Vertical Rectus Abdominis Myocutaneous Flap in the Management of Complex Periprosthetic Joint Infection of the Hip"

_microorganisms, 2025, doi:10.3390/microorganisms13081962_

Round 1
Reviewer 1 Report
Comments and Suggestions for Authors
This article addresses the important and clinically challenging problem of persistent periprosthetic hip infections. The authors present a small case series (n=5) in which VRAM flap reconstruction was used as the definitive treatment option. Despite the limitations typical of retrospective studies and small sample sizes, the text offers interesting clinical observations. Despite its limitations (small sample size and retrospective nature), the article provides a valuable contribution to the treatment of PJI and demonstrates the practical application of microsurgical techniques in challenging clinical cases. The authors accurately identify a niche in which VRAM flap use can be a valuable option. The clinical history, comorbidities, microbial etiology, treatment course, and complications are described. The collaboration between orthopedic surgeons, plastic surgeons, and infectious disease specialists enhances the credibility of the presented therapeutic approach. The authors present compelling data regarding the use of VRAM flaps in the treatment of periprosthetic hip infections. The results are presented clearly, and the discussion aptly places them in the context of the literature. The emphasis on flap survival and healing, as well as the honest identification of the study's limitations, are valuable. The comparison to other reconstructive techniques is also noteworthy, although in some areas more detailed numerical data and standardized functional assessment are lacking. A more in-depth analysis of long-term donor-site complications is also worth considering. For patients with complicated PJI, functional data and quality of life assessment are also important. If these were not collected, this should be noted as a limitation. Donor-site hernias and necrosis are mentioned, but there is no in-depth analysis of their impact on the patient's quality of life. A brief discussion of the impact of these complications on subsequent mobility, comfort, and the potential need for further procedures would improve the clinical picture of the therapy.
Author Response
We thank the reviewer for these valuable observations regarding the importance of long-term functional outcomes and donor-site morbidity following muscle flap reconstruction for complex PJI.
In response, as noted by Suda et al., the lack of standardized functional scoring systems in this patient population limits our ability to draw robust comparisons across studies. We have acknowledged this gap and its impact on the generalizability of existing data, including our own (see lines 323-324). We believe that this underscores the need for future prospective studies incorporating validated tools to assess function and quality of life in this high-risk cohort.
Additionally, we have expanded our discussion of donor-site complications in greater detail, referencing specific complications such as ventral hernias, graft edge necrosis, and seroma formation. While these complications were managed non-operatively in our cohort, we agree they represent important considerations in the overall clinical picture. We have added commentary on their potential impact on postoperative mobility, comfort, and the need for further procedures, and highlighted the absence of formal quality-of-life metrics as a limitation of the current study (see lines 315-320).
We hope these additions strengthen the manuscript and provide a more complete understanding of the clinical implications of the VRAM flap in managing complex hip PJIs.
Reviewer 2 Report
Comments and Suggestions for Authors
This manuscript is a retrospective case series study to explore the application effect of the VRAM flap in complex multibacterial hip PJI. The research topic has clear clinical significance and focuses on the current treatment difficulty - multi-bacterial infection combined with soft tissue defects. The author proposed the VRAM flap as a solution and analyzed it in combination with actual cases, which is novel and of practical value.
It is recommended that all strain names be uniformly expressed in italicized Latin in accordance with the international journal practice.
In the results section, it is advisable to consider supplementing the analysis of the changing trends of the bacterial spectrum before and after the operation, for example: In which cases did the bacterial spectrum change from a single species to multiple species? Are there any newly added drug-resistant bacteria? Is it related to open wounds or previous negative pressure therapy?
Author Response
We appreciate the reviewer’s observation regarding the proper formatting of bacterial strain names. We have carefully reviewed the manuscript and revised all microbial species names to follow international conventions, expressing them in italicized Latin throughout the text, tables, and figure legends where applicable.
The paragraph in the Discussion section (see lines 299–306) highlights the microbiological evolution observed in our cohort. As shown in Table 2, the initial organisms were identified after the first surgical intervention for PJI, while subsequent debridements revealed the presence of additional pathogens. Unfortunately, all patients progressed to polymicrobial infections during the course of treatment. This trend likely reflects secondary colonization associated with chronic open wounds and repeated interventions.
Our findings align with those of Marculescu and Cantey, who demonstrated a higher likelihood of polymicrobial infections in patients with persistent wound drainage or soft tissue compromise. Other studies have similarly reported elevated rates of polymicrobial colonization in hip PJI cases managed with muscle flaps. These observations reinforce the need for early soft tissue closure to limit microbial diversity and reduce the risk of treatment failure.
We would like to note, however, that the primary focus of this study was on the surgical aspect of PJI management using the VRAM flap. Therefore, we did not pursue a comprehensive microbiological trend analysis.
Reviewer 3 Report
Comments and Suggestions for Authors
The paper “A single tertiary-care center experience using vertical rectus abdominis myocutaneous flap in the management of complex periprosthetic joint infection of the hip” is devoted to the study of the use of a vertical rectus abdominis myocutaneous flap in the case of periprosthetic joint infection of the hip joint.
The significant risks of prosthetic joint infection necessitate multiple surgeries. This results in soft tissue loss and fibrosis. The vertical rectus abdominis myocutaneous flap is one of the proven techniques, however, the studies conducted do not address its impact on the treatment of periprosthetic hip infection. The data of 12 initially selected patients were used for the study, six of whom were found to have tumors and one case of infection secondary to an intertrochanteric femoral fracture. After their exclusion, 5 patients were selected. The observation was carried out for 28 months. In 80% of cases, wound healing and freedom from infection were achieved, and in 20% of cases, permanent sinus drainage with preservation of the implant was observed. As a result, the survival rate of the flap was 100%. There were no major complications requiring reoperation. Reconstruction with a vertical rectus abdominis myocutaneous flap is a reliable option for the treatment of periprosthetic hip infection. The infection was controlled, but suppressive antibiotics were required.
There are some points in the work that require corrections:
- Is using data from just five patients enough?
- What is the percentage of complications with local flaps compared to vertical rectus abdominis myocutaneous flap? What is the survival rate of other types of flaps?
Author Response
We thank the reviewer for this important observation. We acknowledge that the small sample size is a limitation of this study. However, the rarity and complexity of persistent, polymicrobial periprosthetic joint infection (PJI) of the hip with associated soft tissue defects requiring flap coverage makes this a challenging clinical scenario with limited available data in the literature. This study represents one of the first case series specifically evaluating the use of the vertical rectus abdominis myocutaneous (VRAM) flap in this context. The detailed description of surgical technique, patient selection, outcomes, and complications provides meaningful insight and may serve as a foundation for future multicenter studies or larger cohort analyses. This limitation is now explicitly acknowledged in the Discussion section (lines 321–324).
Reviewer 4 Report
Comments and Suggestions for Authors
This is a highly interesting and valuable manuscript addressing a clinically challenging issue: the management of complex periprosthetic joint infections (PJIs) of the hip. However, in its current form, the manuscript appears more suitable for an orthopedic journal than for a microbiology-focused publication. It includes a substantial amount of surgical detail but relatively limited information on clinical microbiology, infection diagnostics, and anti-infective treatment. This represents a significant limitation given the scope and readership of the journal.
Specific recommendations:
The Introduction would benefit from an additional paragraph discussing the etiology and pathogenesis of PJI with emphasis on biofilm formation.
The Materials and Methods section should be expanded to include a more detailed description of the microbiological diagnostic procedures, including sampling methods, the number of intraoperative cultures, criteria for interpreting results, and a clear distinction between colonization and infection.
In the Results section, a dedicated subsection titled “Microbiological findings and antibiotic therapy” should be added. It should provide:
A list of identified microorganisms, distinguishing pathogens from colonizers, along with any relevant resistance mechanisms;
A summary of the antibiotic treatment strategies (agents used, rationale for selection, duration of therapy);
An outline of suppressive antibiotic therapy, including indications, drugs used, and treatment duration.
The Discussion should place greater emphasis on microbiological aspects, particularly in the context of treatment challenges and pathogen-related outcomes.
Comments on manuscript structure - sections “Materials and Methods” and “Results”:
The current structure of sections 2-6 leads to a mixing of methods and results, which impairs the clarity of the manuscript.
Section 3 (“Outcomes”) contains only endpoint definitions and should be moved to “Materials and Methods” as a subsection titled “Outcome definitions.”
Section 4 (“Surgical History and Infection Characteristics”) combines clinical/microbiological data with results. This entire section should be relocated to the “Results” section and incorporated as the first subsection (e.g., “Patient characteristics and surgical history”).
Section 5 (“Surgical Technique”), while informative, belongs in the “Materials and Methods” section as a separate subsection. It should also be condensed, as the current level of detail is more appropriate for a surgical or orthopedic journal.
Section 6 (“Results”) presents fragmented data. Some information (e.g., surgical history) has already been described in Section 4. For narrative consistency and clarity, I recommend consolidating all findings into a single “Results” section with the following suggested subsections:
Patient characteristics and surgical history
Surgical outcomes
Complications and functional outcomes
Microbiological findings and antibiotic therapy (new subsection).
Author Response
1–5. Microbiology and antibiotic treatment content:
We respectfully note that this study was not designed to explore microbiological diagnostics or antibiotic strategies in depth. Our objective was to examine the surgical application of the VRAM flap, and as such, we have intentionally limited the microbiological discussion to general context and trends. As highlighted in the Discussion (lines 299–306), we do acknowledge the progression to polymicrobial infections and reference existing literature supporting this phenomenon, particularly in the context of chronic wounds and open management. However, we believe a more detailed microbiological analysis—including resistance mechanisms, culture interpretation, and antibiotic protocols—is beyond the scope of this surgical study and may be more appropriate for future work specifically focused on infection.
6.Comments on manuscript structure:
We appreciate the reviewer’s suggestions regarding manuscript clarity and have revised the structure accordingly:
- “Outcome definitions” have been moved to the Materials and Methods section.
- Surgical history and patient characteristics have been consolidated into the Results section.
- The “Surgical Technique” subsection has been relocated to Materials and Methods and moderately condensed to improve focus.
Round 2
Reviewer 4 Report
Comments and Suggestions for Authors
I appreciate the editorial improvements made to the manuscript’s structure, which have enhanced its clarity and organization.
However, the authors have not addressed the core concerns raised in my initial review-particularly regarding the limited microbiological content. As the manuscript is intended for a microbiology-focused journal, microbiological aspects are essential and should not be overlooked.
I understand that the authors consider these elements to be beyond the scope of their surgical study. However, this reinforces the concern that the current version of the manuscript may not be well aligned with the focus and readership of this journal.
Unless the authors are able to incorporate a more detailed microbiological analysis and discussion, in line with my initial review, I would recommend considering submission to a surgical or orthopedic journal, where the strengths of the manuscript may be more appropriately appreciated.
Author Response
We appreciate the recognition of the editorial improvements to the manuscript’s structure and clarity. In response to the reviewer’s core concern regarding the limited microbiological content, we have now incorporated several additions to strengthen this aspect of the manuscript:
-
The Materials and Methods section has been expanded to include details of microbiological sampling methods.
-
A detailed table summarizing the identified organisms, resistance profiles, and antibiotic regimens—including suppressive therapy—is provided to support the microbiological analysis presented in this study (Table 4).
-
A new subsection titled "Microbiological Findings and Antibiotic Therapy" has been added to the Results. This subsection summarizes:
-
Identified pathogens, including changes over time and progression to polymicrobial infection;
-
General antibiotic strategies employed;
-
The use of suppressive antibiotic therapy, where applicable.
-
We believe these additions offer valuable context for readers interested in the microbiological challenges of persistent periprosthetic joint infections. We hope this enhanced content better aligns the manuscript with the scope of the journal.